Variation in the leaf and root microbiome of sugar maple (Acer saccharum) at an elevational range limit

Wallace Jessica 1
Laforest-Lapointe Isabelle 1 2
Kembel Steven W. kembel.steven_w@uqam.ca 1
1 Département des sciences biologiques, Université du Québec à Montréal , Montréal , Québec , Canada
2 Departments of Physiology and Pharmacology, and Pediatrics, University of Calgary , Calgary , Alberta , Canada
Amend Anthony
Electronic publication date: 2018 Aug 14
Publication date: 2018
Volume: 6
Electronic Location ID: e5293
Received 2018 Feb 12; Accepted 2018 Jul 2
Copyright: ©2018 Wallace et al.
Copyright year: 2018
Copyright holder: Wallace et al.
License: This is an open access article distributed under the terms of the Creative Commons Attribution License, which permits unrestricted use, distribution, reproduction and adaptation in any medium and for any purpose provided that it is properly attributed. For attribution, the original author(s), title, publication source (PeerJ) and either DOI or URL of the article must be cited.
License URL: https://creativecommons.org/licenses/by/4.0/

Keywords: Plant-microbe interactions, Sugar maple, Forest ecology, Environmental gradient, Microbial ecology, Range limit, Phyllosphere, Rhizosphere, Endophyte, Epiphyte

Funding: Natural Sciences and Engineering Research Council of Canada (NSERC) Fonds de Recherche du Québec - Nature et Technologies (FRQNT) Canada Research Chairs Program Financial support was provided by the Natural Sciences and Engineering Research Council of Canada (NSERC), the Fonds de Recherche du Québec - Nature et Technologies (FRQNT), and by the Canada Research Chairs Program. The funders had no role in study design, data collection and analysis, decision to publish, or preparation of the manuscript.

==============================
Background

Bacteria, archaea, viruses and fungi live in various plant compartments including leaves and roots. These plant-associated microbial communities have many effects on host fitness and function. Global climate change is impacting plant species distributions, a phenomenon that will affect plant-microbe interactions both directly and indirectly. In order to predict plant responses to global climate change, it will be crucial to improve our understanding of plant-microbe interactions within and at the edge of plant species natural ranges. While microbes affect their hosts, in turn the plant’s attributes and the surrounding environment drive the structure and assembly of the microbial communities themselves. However, the patterns and dynamics of these interactions and their causes are poorly understood.

Methods

In this study, we quantified the microbial communities of the leaves and roots of seedlings of the deciduous tree species sugar maple (Acer saccharum Marshall) within its natural range and at the species’ elevational range limit at Mont-Mégantic, Quebec. Using high-throughput DNA sequencing, we quantified the bacterial and fungal community structure in four plant compartments: the epiphytes and endophytes of leaves and roots. We also quantified endophytic fungal communities in roots.

Results

The bacterial and fungal communities of A. saccharum seedlings differ across elevational range limits for all four plant compartments. Distinct microbial communities colonize each compartment, although the microbial communities inside a plant’s structure (endophytes) were found to be a subset of the communities found outside the plant’s structure (epiphytes). Plant-associated bacterial communities were dominated by the phyla Proteobacteria, Acidobacteria, Actinobacteria and Bacteroidetes while the main fungal taxa present were Ascomycota.

Discussion

We demonstrate that microbial communities associated with sugar maple seedlings at the edge of the species’ elevational range differ from those within the natural range. Variation in microbial communities differed among plant components, suggesting the importance of each compartment’s exposure to changes in biotic and abiotic conditions in determining variability in community structure. These findings provide a greater understanding of the ecological processes driving the structure and diversity of plant-associated microbial communities within and at the edge of a plant species range, and suggest the potential for biotic interactions between plants and their associated microbiota to influence the dynamics of plant range edge boundaries and responses to global change.

Introduction

Microorganisms such as bacteria and fungi inhabit all parts of terrestrial plants including the leaf and root compartments (Andrews & Harris, 2000). The microbial communities that inhabit these plant structures have many beneficial effects on the host’s functions including protecting against pathogens (Innerebner, Knief & Vorholt, 2011), synthesizing growth hormones (Gourion, Rossignol & Vorholt, 2006) and providing nutrients (Davison, 1988). The leaf and root compartments can be colonized inside (endophytes) and outside (epiphytes) the plant’s structure (Vorholt, 2012). These plant-associated microbial communities harbour great biodiversity both on the leaves (Lambais et al., 2006) and roots (Lundberg et al., 2012). The dynamics, interactions and biodiversity of these microbial communities as well as the role and functions of most of the microbial species present are poorly understood. In recent years, advances in environmental DNA sequencing technologies have allowed us to investigate and quantify the structure of bacterial and fungal communities and examine the driving factors behind their ecology and variation. Studies have found that microbial communities are influenced by host species (Redford et al., 2010; Kembel et al., 2014; Laforest-Lapointe, Messier & Kembel, 2016), anthropological modifications of the environment (Sieber, 1989) and host genotype (Bulgarelli et al., 2012) among other factors, and distinct communities occur in different plant compartments (Edwards et al., 2015). However, there are relatively few studies that have investigated the microbial communities from both above and belowground compartments of a single plant species (but see Lambais, Lucheta & Crowley, 2014; Bai et al., 2015; Wagner et al., 2016), and we know little about how selective pressures and neutral evolutionary processes influence plant-microbe interactions along an environmental gradient.

Changes in global climate are affecting plant ranges, allowing some species to increase their ranges while others are facing range contraction or extinction (Morin, Viner & Chuine, 2008). It is expected that the plant-associated microbial community structure will also be affected by changes in the biotic and abiotic environment (O’Brien & Lindow, 1988) because of host phenotype plasticity, as demonstrated for the leaf fungal communities of the European beech (Fagus sylvatica; Cordier et al., 2012). Recent experiments have shown that host water deficiency drives plant microbiome through changes in host phenotype (Naylor et al., 2017; Santos-Medellín et al., 2017; Fitzpatrick et al., 2018). However, Fierer et al. (2011) showed that bacterial diversity of leaf surfaces, of the organic soil, and of the mineral soil did not change across an elevational gradient, suggesting that changes in abiotic and biotic conditions might not always be a limiting factor for bacterial diversity. The role of biotic interactions as a factor in range expansion has been understudied (Van der Putten, Macel & Visser, 2010) and recent research has found evidence suggesting that these interactions may be an important factor in limiting sugar maple range expansion to higher elevations (Brown & Vellend, 2014). In their study, Brown & Vellend (2014) observed that the soil beyond the range limit of the species suppressed sugar maple regeneration, potentially due to antagonistic interactions with fungal pathogens. These results warrant a more profound exploration of the microbial communities within and beyond the sugar maple species range to understand if it is indeed potentially shifts in biotic interactions across the range edge that drive sugar maple survival. As climate change affects the survival of the sugar maple at its southern limit, it will be crucial for foresters to understand the potential importance of plant-microbe interactions at the northern range limits given the economic and ecological importance of this tree species. Although changes in plant-microbe interactions at the sugar maple’s range edge provoked by global warming could have wide repercussions both ecologically and economically, the structure and dynamics of the sugar maple microbiome across elevation gradients is currently unknown.

Sugar maple (Acer saccharum Marshall) is a deciduous tree species native to north-eastern North America where it is an important species both economically (estimated to provide C$200 million in syrup production annually; FPAQ, 2016) and ecologically as one of the dominant trees in temperate forests across eastern North America (Godman, Yawney & Tubbs, 1990; Burns & Honkala, 1990). The species has a latitudinal range from approximately 35°–49°N and is present at low to mid elevations. The species exhibits a range edge both at its latitudinal and elevational limits. At these range limits the fitness of sugar maple trees declines, leading to a sugar maple tree line, and the composition of the forest transitions to dominance by other species. Elevational changes can create a gradient of variation in temperature, moisture and soil attributes even over relatively short distances. These changes affect the growth rate and survival of seedlings in many tree species along these gradients (Sáenz-Romero, Guzmán-Reyna & Rehfeldt, 2006). The upper-elevational range limit of sugar maple is likely to be controlled both by abiotic climatic factors (Siccama, 1974), and changes in biotic factors including herbivory and pathogen damage (Brown & Vellend, 2014). While shifts in biotic factors have been hypothesized to drive the failure of sugar maple to regenerate above its elevational range limit (Brown & Vellend, 2014), the structure and diversity of the sugar maple microbiome has not been quantified at this range limit.

Here, our main aim is to characterize the structure and diversity of microbial communities found on the deciduous tree species sugar maple (Acer saccharum) at Mont-Mégantic, Quebec, where a distinct sugar maple tree line occurs between 600 and 1,070 m above sea level (m.a.s.l.; Brown & Vellend, 2014). At this site sugar maple is a dominant canopy species of the deciduous forest below this elevational range limit, but the forest transitions into spruce (Picea spp.) and balsam fir (Abies balsamea (L.) Mill.) canopy dominance at higher elevations. It has been predicted that as the climate warms, sugar maples will expand their range north (Goldblum & Rigg, 2005; Graignic, Tremblay & Bergeron, 2014) while decreasing abundance in the southern populations (Iverson et al., 2008). Sugar maple seeds at the northern range edge of the species distribution have the highest seed germination percentage, suggesting that genetic and phenotypic changes will interact to influence sugar maple fitness in response to climate change (Solarik et al., 2016). However, variation in plant-microbe interactions might also influence plant species’ fitness under changing environmental pressures. Due to the ecological and economical importance of potential shifts in the distribution of these species, the aim of our study is to investigate how the plant leaf and root microbial communities differ within versus at the edge of its natural distribution.

In this study, our objectives were (1) to quantify the microbiome of sugar maple seedlings by comparing bacterial communities among four different plant compartments (leaf endophytes and epiphytes, and root endophytes and epiphytes) and describing the root endophyte fungal communities, (2) to test whether microbial community structure changes at the sugar maple elevational range limit, and (3) to understand if these changes in microbial community structure across an elevational range limit are consistent across different plant compartments. We hypothesized that distinct microbial communities inhabit each plant host compartment and that compartments will differ in their response to elevational gradients since each compartment represents a unique habitat in terms of exposure to abiotic and biotic conditions, therefore imposing a selective pressure on the local microbial pool. We furthermore hypothesized that the diversity of plant-associated microbial communities should decrease along the gradient from epiphytic to endophytic communities for both leaves and roots (e.g., Bodenhausen, Horton & Bergelson, 2013). We also expected to find a higher diversity in belowground compartments compared to aboveground compartments due to the high abundance of microbes present in soils and on plant roots (e.g., Berendsen, Pieterse & Bakker, 2012).

Materials and Methods

Specimen collection

Naturally regenerating Acer saccharum seedlings were collected in July 2013 from the eastern slope of Parc National du Mont-Mégantic, Quebec, Canada (45°26′51″N, 71°06′52″W). Ten seedlings were randomly selected and collected from each of four sites for a total of 40 seedlings (Table S1). All seedlings were under 10 cm in height and between the ages of two to seven years. The first two sites occurred at the species’ range edge (further on referred to as the “edge” elevation in tables and figures), between 790 and 830 m.a.s.l, where the sugar maple tree line occurs and the forest transitions into balsam fir dominated stands. The other two sites occurred within the sugar maple’s natural range (further on referred to as the “within” elevation) between 720–750 m.a.s.l, and located just below the tree line where sugar maple dominates the stands (Fig. S1). Within each elevational band, the two collection sites were separated by approximately 1 km. All samples were immediately placed in sterile roll bags, transported on ice within the day to the lab, and frozen at −80 °C until processing.

Sample preparation and DNA extraction

We collected the microbial communities from four compartments of each seedling: leaf epiphytes (phyllosphere), leaf endophytes, root epiphytes (rhizosphere), and root endophytes. The exterior surfaces were the rhizosphere, defined as the surface of the roots and the soil within 1 mm from the roots (Clark, 1949), and the phyllosphere, defined as the aboveground leaf surfaces of the plant (Ruinin, 1965). The root and leaf tissues were first separated from one another at the base of the seedling stem. All seedlings had two leaves and had reached a similar growth stage at the time of collection.

In a separate procedure for each compartment, the epiphytic microbial communities of the root tissues and leaf tissues were removed respectively with a 5-minute agitation wash in 30 mL of 1:50 diluted solution of buffer (1M Tris-HCl, 0.5 M Na EDTA, and 1.2% CTAB) (Kadivar & Stapleton, 2003). The plant tissues were then removed from the buffer solution and the samples were centrifuged at 4,000 rpm for 20 min at 4 °C to form a pellet. The supernatant was discarded and the pellet was transferred to a bead beating tube from the MoBio PowerSoil DNA Isolation kit (Carlsbad, CA, USA), following the standard protocol for this kit with the exception that the samples were vortexed for 15 min instead of 10.

The endophyte communities were then processed separately through a series of steps in order to first remove all remaining epiphytic bacteria and fungi. The following steps describe how, after the surface of roots and leaves were sterilized and washed to remove all remaining microbial cells, we finely sectioned the tissues and agitated them in a bead beating tube to release as many endophytic microbes as possible. We started by doing a first wash to ensure no epiphytes were still present by placing the tissues in 30 mL of ethanol and vortexing the tubes for 5 min. The ethanol was then removed and the samples were washed three times with DNA-free water for 3 min. After washing the tissues, the samples were finely sectioned and transferred to a bead beating tube from the MoBio PowerSoil DNA Isolation kit (Carlsbad, CA, USA). The protocol was followed with the exception that the samples were vortexed for 45 min instead of 10 to ensure the plant tissues were lysed to improve endophyte DNA yield. The isolated DNA samples were frozen at −80 °C until further processing.

PCR and multiplexing for 16S rRNA gene sequencing

The samples were amplified and barcoded using a two-step PCR process to prepare them for Illumina sequencing following the protocol described by Kembel et al. (2014). Although this protocol has been used for many studies (i.e., Kozich et al., 2013; Fadrosh et al., 2014; Kembel et al., 2014; Kembel & Mueller, 2014) to reduce the number of primers while maintaining the diversity of unique identifiers (Gloor et al., 2010), we acknowledge that this method could also potentially increase the PCR bias since two PCR steps are carried out. The first PCR step used primers which target the V5–V6 region of the bacterial 16S rRNA gene (799F and 1115R Redford et al., 2010). The primers exclude cyanobacteria in order to exclude plant chloroplast DNA. These primers are modified with a 5′ tail which adds a 6-bp barcode and partial Illumina adaptor sequence to the 16S fragments during PCR (modified 799F: 5′- CGGTCTCGGCATTCCTGCTGAACCGCTCTTCCGATCT xxxxxx AACMGGATTAGATACCCKG; modified 1115R: 5′-ACACTCTTTCCCTACACGACGCTCTTCCGATCT xxxxxx AGGGTTGCGCTCGTTG, where “x” represents barcode nucleotides).

Twenty-five µL PCR reactions were run containing 5 µL 5xHF buffer (Thermo Scientific, Waltham, MA, USA), 0.5 µL dNTPs (10 µM), 0.5 µL forward primer (10 µM), 0.5 µL reverse primer (10 µM), 0.25 µL Phusion Hot Start II polymerase (Thermo Scientific, Waltham, MA, USA), 4 µL of genomic DNA, and 14.25 µL molecular-grade water. The reaction was performed using: 30 s initial denaturation at 98 °C, 20 cycles of 10 s at 98 °C, 30 s at 64 °C, and 30 s at 72 °C, with a final 10-minute elongation at 72 °C. This was performed in triplicate for each sample and the products were pooled and cleaned using the Bio Basic EZ-10 Spin Column kit (Markham, Ontario, Canada) and resuspended in 40 µL of solution elution buffer. The second stage of the PCR amplification was performed using this first stage PCR product as a template. The primers used were custom HPLC-cleaned primers to further amplify 16S products and complete the Illumina sequencing construct (PCRII_for: 5 ′-AAGCAGAAGACGGCATACGAGATCGGTCTCGGCATTCCTGC ;PCRII_rev: 5′-ATGATACGGCGACCACCGAGATCTACACTCTTTCCCTACACGACG).

Single reactions were run for each sample with the same reagents and conditions as the first PCR step with the exception that the cycle amount was changed to 15 instead of 20. A ∼445-bp fragment was isolated by electrophoresis in a 2% agarose gel and DNA was recovered with the Bio Basic EZ-10 Spin Column kit. A multiplexed 16S library was prepared by adding equimolar concentrations of DNA from each sample. The resulting DNA library was sequenced on an Illumina MiSeq 250-bp paired-end sequencing platform at the University of Montreal, Quebec.

PCR and multiplexing for ITS fungal sequencing

We used sequencing of the fungal ITS region (Schoch et al., 2012) on environmental DNA samples from the root interior tissues to investigate endophytic fungal communities present in the fine roots of the sugar maple seedlings. Due to budgetary constraints we focused only on fungal root endophytes. The ITS1F primer (Gardes & Bruns, 1993) was chosen as it discriminates against plants (Lindahl et al., 2013). ITS2 (White et al., 1990) was chosen as it shares properties with the ITS1 primer and can obtain similar results (Mello et al., 2011; Bazzicalupo, Bálint & Schmitt, 2013).

The DNA samples were amplified for fungal sequencing using a one-step PCR step and normalization with primers designed to attach a 12-base pair barcode and Illumina adaptor sequence to the fragments during PCR (Fadrosh et al., 2014). The primers amplified the regions ITS1 and ITS2 of the internal transcribed spacer of the nuclear ribosomal coding cistron (Schoch et al., 2012). (ITS1F Forward: 5′- CAAGCAGAAGACGGCATACGAGATGTGACTGGAGTTCAGACGTGTGCTCTTCCGATCT xxxxxxxxxxxx CTTGGTCATTTAGAGGAAGTAA ITS2 Reverse: 5′- AATGATACGGCGACCACCGAGATCTACACTCTTTCCCTACACGACGCTCTTCCGATCT xxxxxxxxxxxx GCTGCGTTCTTCATCGATGC –3′). Where x represents barcode nucleotides.

One 25 µL PCR reaction was run for each sample. This reaction contained 5 µL 5xHF buffer (Thermo Scientific), 0.5 µL dNTPs (10 µM), 0.5 µL forward primer (10 µM), 0.5 µL reverse primer (10 µM), 0.75 µL DMSO, 0.25 µL Phusion Hot Start II polymerase (Thermo Scientific), 1 µL DNA, and 16.5 µL molecular-grade water. The reaction was performed using: 30 s initial denaturation at 98 °C, 35 cycles of 15 s at 98 °C, 30 s at 60 °C, and 30 s at 72 °C, with a final 10-minute elongation at 72 °C. The samples were processed with an Invitrogen Sequalprep PCR Cleanup and Normalization Kit (Frederick, MD, USA) to give all samples a finished concentration of ∼0.55 ng/µl. The samples were pooled with equal amounts and sequenced on the Illumina MiSeq platform at the University of Montreal, Quebec. We included our negative controls in the sequencing run and confirmed that they yielded no sequences therefore confirming the absence of contamination.

DNA sequencing processing and data analysis

Raw sequence data were processed using PEAR (Zhang et al., 2014) and QIIME version 1.8.0 (Caporaso et al., 2010) software using default parameter settings to trim and combine paired-end sequences to single sequences of approximately 300–350 bp in length. Sequences with an average quality score of less than 30 or with a quality window score of less than 5 were trimmed. The reads were de-multiplexed into samples using barcode sequences. This involved combining the forward and reverse barcodes from each combined read into a 12-bp barcode for 16S samples or 24-bp barcode for ITS samples which could then be matched to a sample ID (Hamady et al., 2008).

Chimeric sequences were removed using the Uclust and Usearch 6.1 algorithms (Edgar, 2010). Sequences were then binned into operational taxonomic units (OTUs) at a 97% similarity cut-off rate using Uclust (Edgar, 2010). The OTUs were assigned taxonomy using the Ribosomal Database Project (RDP) classifier (Wang et al., 2007) as implemented in QIIME, with a minimum support threshold of 80% for bacterial OTUs and 50% for fungal OTUs. For 16S bacterial samples each sample was rarefied to 4,500 sequences. This resulted in a total of 116 usable samples from 37 seedlings (Table S2) with 522,000 bacterial sequences. For ITS fungal samples, each sample was rarefied to 10,000 sequences. This resulted in a total of 28 samples from 28 seedlings (Table S2) with 280,000 fungal sequences. Missing samples were due to low sequence read amounts either as a result of extraction, PCR or sequencing errors.

Indicator species analysis

We tested for the significant association of indicator taxonomic groups present using the LDA Effect Size platform (LEfSe) (Segata et al., 2011). LEfSe is a bioinformatics and statistical methodology that couples standard tests for statistical significance with tests encoding biological consistency and effect relevance to identify the features that violate the null hypothesis of no difference between classes (Segata et al., 2011). This tool identifies the subset of features with abundance patterns compatible with an algorithmically encoded biological distribution hypothesis and estimates significant variation size (“effect size”) for each feature using Linear Discriminant Analysis (LDA; Fisher, 1936). This allowed us to compare the compartments in order to identify significant changes in host-microbe relationships and their strength. We compared the bacterial communities up to the genus level in each compartment type of the plant separately with an LDA cut-off of 2. We compared root endophytes versus epiphytes, leaf endophytes versus epiphytes, endophytes from roots versus leaves, and finally the epiphytes from roots versus leaves.

Statistical analysis

We eliminated OTUs from our dataset that were represented by fewer than 20 sequences as this is a commonly used cut off for rare OTUs (Zhan et al., 2014). Data analysis and plotting was performed using the ape (Paradis, Claude & Strimmer, 2004), ggplot2 (Wickham, 2009), picante (Kembel et al., 2010), and vegan (Oksanen et al., 2010) statistical packages for R (R Core Team, 2014). We used the Bray–Curtis, weighted and unweighted UniFrac (Lozupone, Hamady & Knight, 2006) dissimilarity indices to measure variation in the bacterial community structure among plant compartments and between elevations. For the fungal communities, we used the Bray–Curtis dissimilarity values to investigate variation between the root endophyte samples from different elevations. Prior to running PERMANOVAs on community structure we randomly sampled our dataset to obtain a balanced representation of each compartment type and each elevation. We also tested for homoskedasticity of group dispersions using the function betadisper (vegan), a multivariate analogue of Levene’s test for homogeneity of variances. In addition, we included restricted permutations to occur within each of the two sites at each elevation in order to account for spatial variation in bacterial community structure and to test for robustness of the observed patterns at different elevations.

Using Principal Coordinate Analysis (PCoA) ordinations we visualized taxonomic and phylogenetic similarity among plant compartments at the two different elevations. Using the community matrix data of OTU counts, we performed permutational multivariate analysis of variance tests (PERMANOVA; Anderson, 2001) to identify relationships between the microbial communities, elevation and plant compartments. Finally, we measured bacterial and fungal alpha-diversity for each compartment from both elevations using the Shannon diversity index for each community. Because the distribution of bacterial Shannon diversity significantly differed from a normal distribution (Shapiro test of normality, p = 0.1), a Kruskal–Wallis test and a subsequent post-hoc Dunn test were performed to test for differences in diversity from different compartments. We also perform a similar combination of tests for each compartment to measure the change in alpha-diversity across elevations.

To determine whether plant compartments responded similarly to elevation, we evaluated whether changes in Shannon diversity were correlated among plant compartments using correlation tests. We quantified covariation of microbial community structure and diversity among all combinations of compartment types using a Mantel test on Bray–Curtis distances among samples.

The metadata, raw sequences, and R code are available in Figshare as mentioned in the Data Availability section.

Results

Taxonomic composition of bacterial communities

We identified a total of 3,785 bacterial OTUs (sequences binned at a 97% similarity cut-off) from the 116 samples. Our collector’s curve of the number of OTUs per sample reached a plateau, suggesting that we sampled the majority of the bacterial diversity in the sugar maple microbiome (Fig. S2A). An average of 446 ± 17 OTUs (mean ± SE) were found per sample, with 645 ± 16 OTUs per rhizosphere samples, 393 ±26 OTUs per phyllosphere sample, 438 ± 17 OTUs per root endophyte samples, and 206 ± 9 OTUs per leaf endophyte samples. From our data, we detected a core microbiome, a set of microorganisms ubiquitously present across a habitat (Turnbaugh et al., 2007), for each compartment of the plant (leaf endophytes, leaf epiphytes, root endophytes, root epiphytes) as well as across all compartments (whole plant core microbiome). The microbial communities of different compartments contained similar broad taxonomic groups (i.e., phyla) but with high variation in taxon relative abundances among compartments (Table 1). The microbiome of sugar maple including all compartments was composed of four main phyla and 11 major classes. Four of these classes were Proteobacteria (59.4% of sequences): Alpha- (23.1%), Beta- (23.0%), Delta- (2.9%) and Gammaproteobacteria (10.2%). Three of the classes were Acidobacteria (10.6%): DA052 (3.7%), Acidobacteria (3.6%), Solibacteres (3.0%). Three from Bacteroidetes (15.4%): Cytophagia (9%), Saprospirae (3.2%), Sphingobacteria (2.7%). Finally, the phylum and class Actinobacteria (7.8%): Actinobacteria (6.4%) were also abundant (Table 1; Fig. 1A).

Table 1 Relative abundances (%) of the most abundant bacteria phyla and classes associated with sugar maple, for different compartments and from the combined dataset.

Bacterial phyla are represented in bold text while classes are represented in italics.

Taxa	Rhizo.	Root Endo.	Phyllo.	Leaf Endo.	Combined	Taxa is an indicator of:	
Acidobacteria	24.7%	10.4%	2.0%	0.2%	10.6%	–Epiphytes
–Roots	
–Acidobacteriia	9.04%	4.19%	1.43%	0.13%	3.6%	–Roots
–Epiphytes	
–DA052	7.05%	1.63%	0.91%	0.01%	3.7%	–Roots
–Epiphytes	
–Solibacteres	6.28%	2.03%	0.81%	0.02%	3.0%	–Rhizosphere
–Epiphytes	
Actinobacteria	10.4%	16.3%	3.6%	8.6%	7.8%	–Roots	
–Actinobacteria	8.85%	17.52%	4.18%	5.35%	6.4%	–Roots
–Endophytes	
AD3	1.9%	0%	0%	0%	1.1%	–Epiphytes	
Bacteroidetes	9.3%	9.2%	20.5%	20.5%	15.4%	–Leaves	
–Cytophagia	0.43%	0.36%	14.42%	16.48%	9.0%	–Leaves	
–Saprospirae	4.86%	6.43%	1.28%	0.22%	3.2%	–Roots	
–Sphingobacteriia	2.98%	3.29%	3.80%	1.11%	2.7%		
Chloroflexi	3.7%	2.3%	0.3%	0%	1.6%	–Roots	
Proteobacteria	41.9%	55.8%	71.3%	68.9%	59.4%	–Leaves	
–Alpha–	19.3%	22.4%	26.5%	21.9%	23.1%	–Leaves	
–Beta–	7.3%	14.8%	31.1%	40.5%	23.0%	–Endophytes	
–Delta–	5.0%	4.0%	1.9%	1.3%	2.9%	–Epiphytes	
–Gamma–	10.0%	13.9%	11.8%	5.2%	10.2%		
TM7	2.6%	1.1%	0.4%	0.7%	1.0%		

Figure 1 Relative abundances (%) of bacterial (A) and fungal (B) taxa.

(A) shows the different plant compartments of sugar maple seedlings and (B) shows the average for all compartments combined using the samples from within the species’ elevational range.

Figure 2 Cladograms of LEfSe results showing bacterial indicator taxa at the phylum level.

(A–D) show comparison between (A) root epiphytic (green) to endophytic (red) communities; (B) root epiphytic (green) and leaf epiphytic (red) communities; (C) leaf epiphytic (green) and endophytic (red) communities; and (D) leaf endophytic (red) to root endophytic (green) communities. The circles, parentheses, and shading indicate with which compartment the bacterial taxonomic group is significantly associated.

Indicator species analysis of bacterial taxa

Numerous bacterial taxa were associated with different sugar maple plant compartments. We compared the taxa of epiphytic and endophytic communities of each compartment using the LEFse approach and found several associations (Table 1; Figs. 2A and 2C). Then we also compared leaf-associated bacterial communities to the root-associated communities and found that most of the abundant taxa were associated with either leaves or roots (Table 1; Figs. 2B and 2D). We also found several non-dominant bacterial taxa with significant associations with either epiphytic or endophytic communities as well as with leaves or roots (Table S3). We tested whether specific bacterial phyla were associated with the range edge or within range elevations. We analysed each of the four compartments of the plant separately at each elevation. We found that there were many associations with the greatest number occurring in the bacterial communities of the rhizosphere and root endophytes from within the natural range of the sugar maple (Table 2).

Table 2 Bacteria taxa that showed a significant association with sugar maples in either the bacterial communities at species’ range edge (edge) or within species’ range (within) samples using the LDA Effect Size platform (LEfSe).

Taxa	Rhizosphere	Root Endophytes	Phyllosphere	Leaf Endophytes	
Acidobacteria	–	Within	–	–	
Actinobacteria	Edge	–	–	Within	
Armatimonadetes	Edge	Edge	–	–	
Bacteroidetes	–	–	–	–	
Chloroflexi	Within	Within	–	–	
Chlamydiae	–	Within	–	–	
Elusimicrobia	Within	Within	–	–	
Gemmatimonadetes	Within	Within	–	–	
Nitrospirae	–	Within	Within	–	
Planctomycetes	Within	–	–	–	
Proteobacteria	Edge	Edge	–	Edge	
Spirochaetes	Within	–	–	–	
Thermi	–	–	Within	Within	
TM6	–	Within	–	–	
Verrucomicrobia	–	Within	–	–	

Differences in bacterial community structure among plant compartments

Tests using the analysis of variances on the Bray–Curtis dissimilarities were used to investigate variation in bacterial community structure in the different compartments as well as between samples from different elevations. Community structure in replicate sites from the same elevation was not significantly different (Table 3; PERMANOVA; p = 0.374). Each of the four compartments of the plant had a distinct bacterial community structure (Table 3, Fig. 3; PERMANOVA; R2 = 54.7%, blocked on range; p = 0.001). Distinct bacterial communities were also found on seedlings from the elevational range edge versus within the elevational range (Table 3; Fig. 4; R2 = 7.1%, blocked on compartment, p = 0.001) in each of the four bacterial community types.

Table 3 PERMANOVAs on Bray–Curtis dissimilarities and UniFrac distances showing the main drivers of bacterial and fungal community structure.

The models investigate the effect of site identity (model #0, b.comm ∼ site, blocked on elevation), compartment type (model #1, b.comm ∼ elevation/site/type, blocked on elevation), elevation (model #2, b.comm ∼ type/elevation blocked on site), as well as the interaction between elevation, tissue level (root vs. leaf) and subtype (epi- vs. endophytes) (model #3, b.comm ∼ elevation*level*subtype) on bacterial community structure as well as the effect of site identity (model #4, f.comm ∼ site blocked on elevation) and elevation on fungal community structure (model #5, f.comm ∼ elevation/site).

Model	Bray–curtis dissimilarities	UniFrac	
					Unweighted	Weighted	
Bacterial communities	Df	R2 (%)	P-value	R2 (%)	P-value	R2 (%)	P-value	
#0	Site	3	NS	0.374	NS	0.189	NS	0.398	
#1	Type	12	54.7	0.001∗∗∗	31.1	0.001∗∗∗	62.9	0.001∗∗∗	
#2	Elevation	4	7.1	0.004∗∗	7.0	0.002∗∗	6.9	0.003∗∗	
#3	Elevation	1	3.5	0.006∗∗	2.5	0.001∗∗∗	1.8	0.001∗∗∗	
Subtype	1	5.1	0.001∗∗∗	3.9	0.001∗∗∗	6.0	0.001∗∗∗	
Level	1	37.8	0.001∗∗∗	12.9	0.001∗∗∗	45.8	0.001∗∗∗	
Elevation*Subtype	1	1.4	0.064 +	1.6	0.043∗	1.6	0.03∗	
Elevation*Level	1	1.6	0.043∗	1.5	0.054 +	2.4	0.008∗∗	
Subtype*Level	1	3.4	0.004∗∗	2.5	0.003∗∗	2.6	0.004∗∗	
Elevation*Subtype*Level	1	NS	0.112	1.6	0.035∗	NS	0.124	
Fungal communities	Df	R2 (%)	P-value	R2 (%)	P-value	R2 (%)	P-value	
#4	Site	3	NS	0.16	NA	NA	NA	NA	
#5	Elevation	1	13.7	0.001∗∗∗	NA	NA	NA	NA	

Figure 3 Principal Coordinate Analysis (PCoA) on Bray–Curtis dissimilarities of bacterial communities from four different plant compartments.

Permutational analysis of variance (PERMANOVA) blocked on elevation showed significant differences (p = 0.001) among all categories. Colors and shape indicate community identity (root: orange triangles for epiphytes, red squares for endophytes; leaf: turquoise lozenges for epiphytes and green circles for endophytes). Ellipses indicate 95% confidence intervals around samples from each category.

Figure 4 Principal Coordinate Analysis (PCoA) on Bray–Curtis dissimilarities of bacterial (A–D) and fungal (E) communities at sugar maple’s normal elevational range and at elevational range limit.

Colors indicate community identity (root: orange for epiphytes, red for endophytes; leaf: turquoise for epiphytes and green for endophytes). Line type indicates environment type (full line for within-range and dotted line for range edge samples). Permutational analysis of variance (PERMANOVA) showed significant differences between the bacterial communities in all compartments (A–D, p = 0.001, blocked on elevation) and the fungal endophytic communities of the roots (E, p = 0.001). Ellipses indicate 95% confidence intervals around samples from each category.

Covariation of bacterial community structure and diversity among plant compartments

While community composition and diversity differed among compartments, there was significant covariance in composition and diversity among compartments at different elevations. Community composition across elevations (Mantel test on Bray-Curtis distances; Table 4) was significantly and strongly correlated among root endophytes, root epiphytes, and leaf endophytes (r = 0.48 − 0.67, p < 0.001), but more weakly correlated between leaf epiphytes versus endophytes (r = 0.25, p < 0.05), and uncorrelated between leaf epiphytes and root endophytes. The diversity of bacterial communities from different compartments also covaried across elevations (correlation on Shannon diversity; Table 4), with the strongest correlations between the diversity of leaf epiphytes versus endophytes (r = 0.71, p < 0.001) and weaker correlations between root endophytes versus root epiphytes (r = 0.46, p < 0.05). Root fungal endophytes covaried significantly only with the leaf bacterial epiphytes (r = 0.51, p < 0.05).

Table 4 Covariation between (a) microbial community structure (Mantel test on Bray–Curtis dissimilarities); and (b) microbial alpha-diversity (correlation on Shannon indices) among and across compartment types.

(a)	
Compartment	Root endophytes	Root epiphytes	Leaf endophytes	Leaf epiphytes	Root fungal endophytes	
Root endophytes	1					
Root epiphytes	0.56***	1				
Leaf endophytes	0.67***	0.48***	1			
Leaf epiphytes	NS	0.15*	0.25*	1		
Root fungal endophytes	NS	NS	NS	0.18+	1	
(b)	
Compartment	Root endophytes	Root epiphytes	Leaf endophytes	Leaf epiphytes	Root fungal endophytes	
Root endophytes	1					
Root epiphytes	0.46*	1				
Leaf endophytes	NS	0.43+	1			
Leaf epiphytes	NS	NS	0.71***	1		
Root fungal endophytes	NS	NS	NS	0.51*	1	
Notes.

+p < 0.1, ∗p < 0.05, ∗∗p < 0.01, ∗∗∗p < 0.001.

Differences in bacterial community phylogenetic structure

Distinct communities were found between the elevations in the root-associated bacterial communities using PERMANOVA tests on both the weighted and unweighted UniFrac distances (Table 3). Both of the root-associated bacterial communities showed significant variation between the two elevations using both UniFrac distances. The leaf-associated bacterial communities showed a significant difference between the elevations using UniFrac for both the weighted and unweighted index (Table 3; p = 0.002 and p = 0.003 respectively).

Taxonomic composition of fungal communities

Sequencing of fungal root endophytes using the ITS region identified 2044 OTUs from the 28 samples with an average of 258  ± 3 OTUs (mean ± SE) per sample (Fig. S2B). From these 28 seedlings, 18 were from the range edge and contained 952 OTUs, the other 10 seedlings were from within the elevational range and contained 818 OTUs. Taxonomic analysis of fungal communities within the elevational range showed that the most abundant phyla were Ascomycota (40.1%), Basidiomycota (12.4%), and Zygomycota (46.4%) (Fig. 1B). The most abundant Ascomycota classes included Dothideomycetes (7.7%), Eurotiomycetes (2.6%), Leotiomycetes (7.5%), and Sordariomycetes (10.2%). Another abundant class was Agaricomycetes (11.5%) from Basidiomycota (Table 5). Similar to the bacterial communities, replicate plots within each elevation were not significantly different (PERMANOVA; p = 0.16) and were grouped together by elevation for further analysis. Fungal root endophyte communities differed between the within-range and range edge elevations (p = 0.001) (Table 3; Fig. 4E).

Table 5 Relative abundances (%) of the most abundant fungal phyla and classes associated with the root endophytic communities of sugar maples.

Fungal phyla are highlighted in gray and in bold text while classes are represented in italics.

Taxa	Root Endophytes	
Ascomycota	40.1%	
–Dothideomycetes	7.7%	
–Eurotiomycetes	2.6%	
–Leotiomycetes	7.5%	
–Sordariomycetes	10.2%	
Basidiomycota	12.4%	
–Agaricomycetes	11.5%	
Zygomycota	46.4%	

Diversity of bacterial communities

The diversity (Shannon diversity) of bacterial communities of each compartment was compared between elevations using non-parametric kruskal–Wallis tests followed by a post-hoc Dunn test to quantify differences in alpha-diversity. While overall there was no significant difference between elevations for the leaf-associated bacterial communities or the fungal root endophyte communities, there was a significant difference in diversity between elevations for the rhizosphere (p = 0.008) and root endophyte (p < 0.001) bacterial communities (Table 6; Fig. 5).

Table 6 Differences in the diversity of microbial communities of sugar maple compartments at two elevations.

Tests based on Kruskal–Wallis tests followed by post-hoc Dunn tests on Shannon alpha diversity of each compartment at two elevations.

	Chi-squared	Df	P-value	
Bacterial communities				
Rhizosphere	7.040	1	0.008	
Root Endophytes	17.434	1	p < 0.001	
Phyllosphere	0.523	1	0.470	
Leaf Endophytes	1.339	1	0.247	
Fungal communities	
Root Endophytes	4.4471	1	0.035	

Figure 5 Bacterial operational taxonomic unit (OTU) Shannon diversity of sugar maple compartments and post-hoc test of Dunn.

The diversity between compartments of the plant (Rhizosphere, Phyllosphere, Leaf Endophytes, Root Endophytes) was significantly different (p < 0.05) between each pair except phyllosphere and root endophytes. Compartment alpha-diversity was significantly different between the two elevations (pale grey indicates at range’s edge, dark grey indicates within range) only for root endophytes and epiphytes respectively (p < 0.05).

Shannon alpha-diversity for each plant compartment was highest in the rhizosphere and lowest for the leaf endophytes. Root-associated bacterial communities were more diverse compared to leaf-associated communities, and samples from the epiphyte communities (phyllosphere, rhizosphere) were more diverse compared to their respective endophyte communities (Fig. 5). Fungal community diversity of root endophytes also differed significantly between elevations (p = 0.035).

Discussion

Our study characterized the microbiome of different sugar maple compartments within and at the edge of the species’ elevational range, demonstrating that sugar maple-microbe associations are complex and vary across plant compartments and the elevational range limit. The overall taxonomic composition of the different plant compartments was consistent with previous studies of plant and tree species (Davey et al., 2012; Shakya et al., 2013; Kembel et al., 2014). Many abundant bacterial taxa in the sugar maple microbiome were present across all plant compartments but occurred at a higher relative abundance in either leaf or root samples, with some further associated specifically with endophytic or epiphytic habitats (Tables 1– 2; Table S3). The phylum Proteobacteria and the class Alphaproteobacteria were more relatively abundant in the leaf habitat, which concurs with previous studies that found this phylum and class to be dominant in the phyllosphere (Kembel et al., 2014; Laforest-Lapointe, Messier & Kembel, 2016). On the other hand, there were a greater number of taxa with significant associations with the root compartment compared to leaves (Table 2), which could confirm the role of the soil as a consistent reservoir of microbial diversity colonizing the plant rhizosphere and root. The endophyte compartments were less diverse and contained fewer significant associations then their respective epiphyte counterparts, suggesting that there is a filtering process allowing only a subset of the epiphytic taxa to successfully colonize the inside of the plant tissues. These results support previous work showing that plants exert some selection on microbial colonists of their tissues, for example through plant immune signaling (Lebeis et al., 2015).

Each plant compartment was colonized by distinct bacterial communities, and bacterial epiphyte communities found in the rhizosphere were more similar to root endophytes than to leaf communities. Similarly, leaf endophytes were more similar to leaf epiphytes than to the root communities (Fig. 3). The lower diversity of OTUs on sugar maple leaves compared to roots could be explained by the relatively harsh environmental conditions on leaves, which are characterized by UV radiation, low nutrient availability and low moisture (Lindow & Brandl, 2003) while the rhizosphere has relatively high nutrient and moisture availability (Badri et al., 2009; Mendes et al., 2011). However, both endophytic samples showed lower diversity than the epiphytic communities of the same plant compartments. While leaf endophytes have been found to be more diverse than leaf epiphytes (Bodenhausen, Horton & Bergelson, 2013), our data showed the opposite. Our results support a model of community assembly where microbes are progressively filtered as they colonize the plant surface followed by the endophytic compartments (Bulgarelli et al., 2013), with decreases in diversity moving from the epiphytic to endophytic compartments of the plant.

Several bacterial phyla showed a higher relative abundance at a specific elevation, with a higher number of associations occurring within the sugar maple’s elevational range. There was a consistently higher alpha-diversity in the samples within the sugar maple’s elevational range (although this trend was only statistically significant for root endophytes and epiphytes). Conversely, root endophyte fungal communities showed no significant differences in alpha-diversity between elevations. The composition of microbial communities covaried among compartments at different elevations, but these covariances were complex and fungal endophytes covaried with leaf bacterial epiphytes but not with the bacterial communities in other plant compartments. Taken together, these results suggest that bacterial and root endophyte fungal associations with sugar maple may change independently in response to climate change and range shifts. Thus, forecasting the interplay between plant stress responses, plant immune systems, and plant-microbe associations under changing environmental conditions (e.g., Castrillo et al., 2017; Hacquard et al., 2017) may be challenging and difficult to generalize.

Plant-associated microbes influence plant health and fitness (Zamioudis & Pieterse, 2012), resistance to pathogens (Awasthi et al., 2014; Innerebner, Knief & Vorholt, 2011), and ecosystem services such as productivity (Laforest-Lapointe et al., 2017). Biotic interactions with microbial pathogens have been hypothesized to limit the elevational distribution of the sugar maple (Brown & Vellend, 2014), and our study demonstrates there is a shift in plant-microbial associations at this range edge. However, these shifts are complex, with different microbial taxa and plant compartments responding differently to elevation, and their relative importance for the plant host remains unmeasured. Our conclusions are limited by the fact that we cannot distinguish the relative importance of plant phenotype, genotype, and the abiotic and biotic environment to explain these shifts (Edwards et al., 2015; Wagner et al., 2016) and future studies that build upon our results to sample at a broader range of sites and to mechanistically test for the importance of the different taxa we identified using field and greenhouse experiments will be required for a holistic understanding of the importance of the sugar maple microbiome for host fitness and function.

Conclusions

In this study, we used high-throughput DNA sequencing of bacterial and fungal molecular markers to compare the microbial communities of Acer saccharum seedlings from four different plant compartments and along an elevational gradient where a distinct sugar maple tree line occurs. In summary, Acer saccharum seedlings were found to have distinct bacterial communities inhabiting their leaves, roots, and different endophytes compartments. The composition of bacterial and fungal communities associated with sugar maple shifted across the elevational range limit of the species. This study expands our knowledge of the ecology of plant-microbe interactions and the structure and assembly of microbial communities found on sugar maple trees, and suggests several avenues for future work to mechanistically test the importance of plant-microbe interactions along environmental gradients and species range edges.

Supplemental Information

Supplemental Information 1 Supplementary Material

Figures S1–S2 and Tables S1–S3

Click here for additional data file.

We thank Carissa Brown and Mark Vellend for providing seedlings for analysis, and Travis Dawson for assistance in the lab. We thank Briana Whitaker, Naupaka Zimmerman, and an anonymous reviewer for comments that improved the quality of this manuscript.

Additional Information and Declarations

Competing Interests

Author Contributions

DNA Deposition

Data Availability

The authors declare there are no competing interests.

Jessica Wallace conceived and designed the experiments, performed the experiments, analyzed the data, contributed reagents/materials/analysis tools, prepared figures and/or tables, authored or reviewed drafts of the paper, approved the final draft.

Isabelle Laforest-Lapointe analyzed the data, contributed reagents/materials/analysis tools, prepared figures and/or tables, authored or reviewed drafts of the paper, approved the final draft.

Steven W. Kembel conceived and designed the experiments, analyzed the data, contributed reagents/materials/analysis tools, prepared figures and/or tables, authored or reviewed drafts of the paper, approved the final draft.

The following information was supplied regarding the deposition of DNA sequences:

Laforest-Lapointe, Isabelle (2018): 16S ITS DNA raw sequences. figshare. Dataset. https://doi.org/10.6084/m9.figshare.5860092.v1.

The following information was supplied regarding data availability:

Laforest-Lapointe, Isabelle (2018): Megantic_Mapping_File.txt. figshare. Dataset. https://doi.org/10.6084/m9.figshare.6267404.v1

Laforest-Lapointe, Isabelle (2018): ITS_mapping.txt. figshare. Dataset. https://doi.org/10.6084/m9.figshare.6267410.v1

Laforest-Lapointe, Isabelle (2018): Community matrices and metadata. figshare. Dataset. https://doi.org/10.6084/m9.figshare.5859702.v1

Laforest-Lapointe, Isabelle (2018): R Code. figshare. Dataset. https://doi.org/10.6084/m9.figshare.5859381.v1

Laforest-Lapointe, Isabelle (2018): Readme.txt. figshare. Dataset. https://doi.org/10.6084/m9.figshare.6267431.v1.

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
