# Peer review of "Variation in the leaf and root microbiome of sugar maple (Acer saccharum) at an elevational range limit"

_PeerJ, doi:10.7717/peerj.5293_

## Round 0.1 · original submission · Major Revisions

Thank you for your submission. As you'll see, the reviewers were generally enthusiastic about this research, and have suggested some statistical and editorial improvements for a second submission.

Reviewer 1 ·

Basic reporting

I really enjoyed the ecological context introduced in the second paragraph of the introduction but I think it could be better integrated with the objectives of the experiment. For example, why is it ‘expected’ that the plant-associated microbiota will be affected along an environmental gradient (L78-79)? Is it because the microbial species pool will be changing, host phenotype/genotype will be changing, environmental constraints imposed on potential microbial colonists, some combination of explanations? Can you provide some recent work that demonstrates any of these mechanisms? A number of recent publications show how exposure to drought modifies the root microbiome, likely indirectly through changes in host phenotypes (Naylor et al. 2017 ISME, Santos-Medillin 2017 mBIO, Fitzpatrick et al. 2018 PNAS). Could a similar mechanism be operating here?

The authors also allude to the possibility that ‘biotic interactions’ may be contributing to sugar maple range expansion to higher elevation (L82-83). This is very intriguing but I’m craving more here. What biotic interactions? Plant-plant, plant-herbivore, plant-pathogen, plant-mutualist? Are you hinting that plant microbiota may be facilitating host range expansion? How? Are associations with particular microbes beneficial under stressful environments? Again, the papers listed above seem to suggest that under drought plant roots associate with a particular lineage of bacteria, with potential beneficial effects. And how might changes in plant-microbe interactions have widespread ecological and economical effects (L85-86)? Will these altered interactions increase growth, cause evolutionary change, facilitate interactions with other plant species, herbivores, pathogens?

Why were fungal communities only characterized in root endosphere compartments? Throughout the manuscript the fungal portion of the project feels a bit under-explained/justified, an add-on.

Sample preparation and DNA extraction (L130-151). Readers will need more detail either in the main text or the supplement to effectively understand these methods.

L133: ‘aboveground surface of the plant’ – does this include stem, floral, petiole etc? Phyllosphere typically means foliar tissue, excluding the rest of the aboveground tissues.

L135-141: I don’t understand how rhizosphere and phylosphere compartments were separated here. Did you remove all leaves from an individual and vortex these to get the phyllosphere? And then remove all roots and vortex to get the rhizosphere? Did you standardize by # of leaves or roots, or biomass of leaves or roots? Given that there may have been differences in growth between seedlings growing at edge versus within sites was this taken into account at the sampling stage or analytical stage, using biomass as a covariate?

L142-151: I don’t understand the sequence of events in this protocol. I think the authors took samples after epiphyte collection and performed an EtOH wash with water rinses, then finely section them before putting them into DNA extraction tubes. This isn’t clear as written though. Do you have a citation for this particular method of surface sterilizing? Same comment as above, did you standardize the plant biomass going into each extraction across tissues, sites? Also why a 45 min. bead beating stage? Did the authors use a vortex for this step or a tissue homogenizer? 45 mins. in a high powered homogenizer would likely shear all DNA.

Biomarker analysis (L227-237): Given that the LEfSe results are so central to your study I would like to see more description of the modeling process here or in the supplement. A brief description of LDA would be helpful for unfamiliar readers and a lucid description of what ‘effect size’, how it is calculated and statistically tested is needed. With 100s-1000s of microbial taxa tested in analyses like these multiple testing can be problematic. Can the authors comment on how this was controlled for?

L 350: Is there any ecological significance of Mortierella. Given its high abundance maybe it has been documented previously as an important endophyte, mutualist or pathogen?

Experimental design

I’m not familiar with forest ecology in Quebec but stating that 100 m is an elevational gradient is a stretch in any system. The authors state that sugar maple’s elevational range is from 600 – 1000 m, why was more an effort not made to capture the extent of this range? Even 400m seems unlikely to result in any large environmental variation but at least it is biologically relevant for the host. Additionally, given the lack of replication of within and edge populations it’s near impossible to attribute any of the differences observed here to elevation, they could be due to confounding spatial variation. This is a major limitation that needs to be addressed much more openly throughout the manuscript.

Why were soil or other environmental samples accompanying each site not collected? This would have been very informative regarding whether host-associated shifts were caused by a shift in the microbial species pool occurring across sites.

Given that all of the sampling occurred on plant seedlings is there any reason to suspect that results may differ across developmental stages of the plant?

Would it be useful to have an additional biomarker analysis which compares epi- and endophytes separately for roots and leaves? The biomarker analysis using the comparison of endophytes to epiphytes pooled across above and belowground tissues may be missing a number of taxa that differentiate epi and endo habitats in each particular tissue.

One analysis that I would like to see is whether or not measures of diversity and composition covary among epiphytic and endophytic compartments across and within root and leaf compartments, and possibly across elevation. Simple correlation for measures of diversity and a Mantel test for measures of composition could do this.

Figure 4 seems to indicate that the variance in community composition seems to decrease at edge sites in root microbiota. Testing for unequal variance might be informative.

Validity of the findings

L326: According to figure 5, only root endophytic compartments exhibited altered diversity across range. Though Table 5 and figure 5 conflict with respect to sig. of rhizosphere compartments.

L 354: I don’t agree with the biological interpretation of these results. To say that Proteobacteria are more associated with leaves than roots just doesn’t make sense. There are members of the Proteobacteria that are very strongly and exclusively associated with roots and at high abundance (e.g. Rhizobiales). Speaking in terms of relative abundance may be more appropriate here and in other instances regarding the LEfSe results.

L357: I’m not following the logic here. How do greater taxonomic associations found in roots lead to greater effect of the environment on belowground host microbiota?

The authors state in L 360 – 363 (and elsewhere: L381, 385-387, that their results confirm a shift in microbial community structure in leaves and roots across the natural range of sugar maple to its edge. This is much too strongly worded given the inability to actually detect elevation-specific differences in the dataset (see comment in Experimental Design above). Qualifying that the elevational and not geographic range is being considered here is also important as this statement suggests a much larger sampling design than exists (also at L427). Finally, though I appreciate the speculation, that these shifts may be linked to host plant fitness and function is just unfounded. I think if this section were expanded with relevant citations, which demonstrate how shifts in associated microbial communities may be linked to host function, than this would be ok.

In the discussion I was hoping to see more links back to the concepts from the introduction. If we assume that the differences observed here really reflect altered host microbiota across an elevational range, what does this tell us? That the microbial species pool from which host microbiota are assembled has changed? That host phenotype or genotype has changed, thus precipitating a shift in the associated microbiota? Are these changes in host genotype or phenotype being driven by environmental differences across the elevational range, in turn shaping microbial communities (e.g. GXE shown in Edwards et al. 2015 PNAS, Wagner et al. 2016 Nat Comm)? How might changes in associated microbial communities be altering host plant survival or growth across an elevation change?

L404-405: This is misleading. According to figure 5 only root endophytic compartments exhibited statistically different diversity across range.

Fig. 2: What do the light shaded green and red fans mean? Something to do with monophly of association? Why do some clades (e.g. Actinobacteria in panel a) have a fan of one color but individual nodes with a different color?

Fig. 5: no legend for boxplot colors.

Additional comments

Wallace et al. investigate variation in the diversity and composition foliar and root microbiota of Acer saccharum across an elevational gradient. First, the authors should be commended for taking the study of host-associated microbiota out of the lab and into the field. Characterization of host microbiota in natural populations and across environmental gradients is needed for a complete understanding of their ecological importance.

L23-24: Here it sounds like the authors characterized bacterial and fungal communities for each compartment but from reading Table S2, fungal communities were only characterized from the root endophytic compartment? Please clarify.

L52-53: Completely optional, but I’m wondering if the distinction between epi- and endophytic communities should come in now? Here the authors say on the leaves but the study focuses on both in/on leaves and roots.

L62: I completely agree that this is an important question in host microbiome research but I would like to see a bit more explanation here. From an evolutionary perspective, measuring the genetic covariance across traits is important because it can tell us whether pleiotropy and adaptive constraints may exist. I think the same reasoning goes for microbial community structure if you imagine communities across different tissues as ‘traits’, are they controlled through independent mechanisms?

L66: What about sugar maple is ecologically important? Just biomass or something more?

L96: I don’t quite understand how this statement is connected to the current research. Are you suggesting that genetic/phenotypic differences in sugar maple at the expansion front may cause differences in associated microbiota?

L119: Are these trees from an experimental plot or have they recruited to these sites naturally?

Fig S1: The informational content of Figure S1 could be improved. Maybe a Google earth map adjacent to the existing figure that shows the spatial structure of the collection sites? I would like to see information about the spatial separation between sites included in the Specimen Collection section.

L152: Were negative controls used during PCR? Any indication of contamination during the extraction or amplification stage?

L172: Two-step PCR for 16S sequencing seems like it might introduce (even more) bias into the final amplicon pool. Can the authors comment on this?

L202: Just curious why 20 cycles were used for 16S amplification but 35 cycles for ITS amplification? Is this simply due to there being lower fungal DNA yields after extraction?

L265: How were taxa defined as core here? For example, core rhizosphere taxa = found in every rhizosphere, whereas core plant taxa = found in every compartment of every plant?

L331: is the word ‘by’ a typo here?

L360: I don’t understand the ‘associations with planting’ here.

L374-377: This model of microbiome assembly has been proposed. Who knows when it was originally proposed but I usually attribute it to Bulgarelli et al. 2013 AREES.

L421: Here and throughout I think the authors can do a better job drawing from recent advances in host (and particularly plant) microbiome research. We now know that plant immune phenotypes can alter associated microbial communities, that resource availability can trigger immune responses, in turn shaping associated microbial communities (Lebeis et al. 2015 Science, Castrillo et al. 2017 Nature, Hacquard et al. 2017 CHOM). The root microbiome can dynamically respond to abiotic stress, potentially mediating negative effects (Naylor et al. 2017 ISME, Santos-Medillin 2017 mBIO, Fitzpatrick et al. 2018 PNAS).

·

Basic reporting

- The R code available from FigShare appears to be missing the code to perform the PERMANOVA as described in the text at L251. Please add this code to the file.
- The raw data (CSV files, R code, sequences) are available through FigShare, though not referred to in the text or as a section of the manuscript.

Experimental design

- L184, It is unclear why the fungal community sampling and analysis was only done for endophytic roots, when the bacterial sampling was done for four different compartments? Why only sample one component for the fungi, but then make a claim to be broadly comparing bacterial and fungal microbial communities?
- L264-7, I would not consider classes/phyla to be ‘core microbiome’, but rather lower taxonomic levels, such as species or strains. Additionally, for the Biomarker analysis/ISA (L234), it is not clear why only the phyla level was examined. Illumina sequencing produces only amplicon abundances or short genomic regions, it would be worth adding a qualification for the value of this analysis given that finer taxonomic resolution is not being presented and the discussion of variability of groups across elevational ranges and plant compartments does not reflect specific species counts but sequencing reads.

Validity of the findings

No Comments.

Additional comments

Introduction
- L66-101, The two paragraphs beginning at L66 (through to L101), the information regarding the specific biology of Acer saccharum and the more general ecological properties of shifting range limits under climate change with abiotic and biotic interactions are too mixed together. I would suggest a paragraph discussing the more general properties of climate change on species distributions and biotic interactions with the microbiome, and then a more specific paragraph on A. saccharum to increase clarity.
- L83-4, can you describe what this reference found a little more? It seems very relevant and is currently too vaguely stated.

Materials & Methods
- L143-5, it is unclear how this step in obtaining endophytic microbiota is different, (or another step in the process?), from those steps described at L146-9. What is meant by “the interior samples”?
- L186, do you mean ITS2 primer, or ITS2 region. Please clarify. Similarly, at L192, do you mean ITS1 and ITS2 primers, or both regions of the ITS?
- L236, it is confusing to switch between the terms ‘interior’/endophytic vs. ‘exterior’/epiphytic. Please clearly define interior/exterior in the context of compartments earlier in the Methods.

Results
- L276, This header does not match with the terminology that was used previously, “indicator species analysis” needs to be defined. Previously, the terms ‘indicator taxonomic groups’ and ‘biomarkers’ were used (L227-9).
- L320, I think you mean Fig. 4e
- L333-5, The results presented in this line match the corresponding figure 5, but do not make sense in light of the statement presented at L327. How are these results different, does the bacterial diversity differ among the elevations for the rhizosphere or not?

Discussion
- L351, this might
- L390-4, But how would this model of experimentation be linked to the effects of the microbiota on plant fitness rather than just the effects of the abiotic environment (re-climate change) on plant fitness?
- L406-9, You can’t say this for all fungal communities, but only root endophytes which is all that you measured. Please adjust text to indicate this qualification.

Figures
- The description for Figure 2 is not very clear, please revise to make it clearer that each figure shows a dichotomy in how the different taxa respond to endo/epi or leaf/root.
- Fig.2B has the opposing color scheme from Fig.3 for root vs. leaf.

·

Basic reporting

This is a well-written manuscript describing a study on plant-associated bacterial and fungal communities associated with Acer saccharum (sugar maple) within and at the edge of its elevational range in Quebec. The authors use a high-throughput culture-free MiSeq-based approach to quantify microbial community composition and diversity. They compare the bacterial communities between leaf surface and interiors, root surfaces and interiors, and the fungal communities from within roots, between two sites within and two sites at the upper edge of the elevational range. They find differences between the plant compartments and between plants sampled within and at the edge of the elevational range.

The literature is, for the most part, appropriately used to provide context and is cited appropriately. I was surprised not to see mention of the Fierer et al. study from 2011 (published in Ecology) on elevational ranges of phyllosphere microbial communities, since it seems quite relevant. I've included the citation below. The figures and tables are professionally done, except for a few omissions related to the captions or legends (detailed below).

The raw sequence data are available, however they may be incomplete. The barcode sequences are not provided for 16S or ITS, nor the metadata necessary to demultiplex the raw fastq files, so the raw sequencing data are currently unusable. A quick grep search of the fastq files suggests that they contain just the sequences themselves, and lack the 16S or ITS primers and the barcodes that would be necessary for de-multiplexing them. This information does not seem to be encoded in the fastq sequence names, either, as far as I can tell.

Additionally, there is no mention in the text of where the raw data will be deposited (even a placeholder), so it is not clear that they will be placed into an appropriate publicly available sequence repository (e.g. NCBI SRA) upon publication.

There were a number of minor technical errors in the methods that I point out below.

Experimental design

In my opinion, the research questions inherent in this study (the extent to which microbial communities vary within plants and within and at thresholds of a plant's elevational range) are important ones where additional work is welcome. I agree with the authors that only a few (although the number is growing) plant microbiome studies simultaneously assess the communities in multiple plant compartments. I also think that the question of whether the microbial communities associated with plant hosts change at the edge of the host's elevational range is an interesting one and is worth pursuing.

I would have liked to see some information about the geographic proximity of the two within and two edge sites. My understanding of how the statistical analyses were presented in the paper is that all of the samples were treated independently. However, because there were two sites in each of the major classes (within and at the edge) there may be pseudoreplication within a given site that is structuring the data and needs to be addressed.

The other study design concern I had was in relation to the age of the saplings. From looking at the metadata table provided as a supplement, it looks like all of the saplings sampled from the range edge sites are two years old, while the ones from within the range varied from 2-7 years. How can you be sure that the effects you observed are due to the differences in sapling location (within vs edge) and not just due to differences in plant age? It seems like an additional test on the within range data specifically focusing on sapling age could be used to test for an effect. If there is none, then this should be stated in the results.

The authors incorrectly state that they used the ITS1 primer for their fungal sequencing. The primer they list out fully in the text is ITS1F, the fungal-specific version of ITS1 that was first presented in Gardes and Bruns 1993. These are different primers; the F isn't just for 'forward'.

I liked that the field sampling permit was included.

Validity of the findings

While I have confidence for the most part in the wet lab components of this project (aside from no mention of a negative extraction or PCR controls, and no citation for the effectiveness of the surface sterilization method used -- most studies I've read use a dilute bleach solution in addition to ethanol to remove surface microbes and degrade their DNA), I have several fairly major concerns with the statistical analyses that will definitely need to be adressed before acceptance. The most important one is that I am not convinced that the permutational p values that were presented in association with the PERMANOVA tests are meaningful. In many of the cases where the authors are making comparisons between plant compartments or sites, they would need to block their permutations to only permute within a given plot, compartment, or elevation. Otherwise, treating all samples as independent ignores the structure in the sampling design and could serve to inappropriately lower the p-values derived from the calculations. The authors mention on line 293 that communities within a given elevation (within vs edge) did not show significant between-plot differences, but it is unclear which samples they are talking about here (which compartment, bacteria vs fungi, etc). To analyze differences in compartments, the permutations should be blocked by elevation, otherwise you risk confounding elevation and compartment effects. To analyze differences in elevation, permutation of samples should be blocked by compartment.

These nested permutation designs can be specified with the permute package in R, and then these restricted permutations can be applied to tests such as the PERMANOVA (via the adonis or adonis2 functions in the vegan package). However, these restricted permutations require a balanced experimental design, so in order to use this approach, the authors would need to subsample their data to the least number of samples in a given category, and this will lower their statistical power. It may be that after recalculating the proper p values, the results all remain valid, but I think it is important to correct this before the manuscript is accepted for publication. They should also test for multivariate homoskedasticity, as big differences in multivariate dispersion (such as those shown in the NMDS ordination plots) can influence the conclusions from a PERMANOVA analysis. The authors should also check and verify that the assumptions of the parametric ANOVA (independence, homoskedasticity, normality of the residuals) are met for their Shannon diversity comparisons, or else switch to a test that is robust to non-parametric data.

While I appreciate the authors including the R script they used for their analysis, it seems to be incomplete -- there is no portion of the code that runs the PERMANOVA tests and so it is impossible to assess if these were done correctly or not.

On a similar note, I think it would be appropriate to include the commands and parameters used for the PEAR and QIIME processing of the data. These bioinformatic pipelines have many parameters, each of which can exert a large influence on the final outcome and conclusions of a study and are not otherwise mentioned in the methods as described.

I appreciate the inclusion of the OTU, taxonomy, and metadata tables as supplementary info, but I think the metadata tables in particular could be improved with the addition of the units used in each of the numeric columns and some sort of key (perhaps in a README) to interpret the codes used in the categorical ones.

Additional comments

The wording in the Methods section of the abstract is imprecise. You did not quantify 'bacterial and fungal community structure in four plant compartments' -- you only did this for the bacteria. I'd suggest revising to make this more clear. This is also true in the Results -- you don't show that the communties differ in all four compartments for both bacteria and fungi, only bacterial data are presented in all four plant compartments. You don't have the data to show that 'the fungal genus Mortierella was also very abundant in all compartments'. Upon reading these I thought that perhaps I was misunderstanding the study design, but table S2 clearly suggests that the only compartment you sampled for fungi was the root interior.

I am not clear why only the fungi from the root interior were sequenced. I think the rationale should be included -- this is particularly confusing because in several places in the abstract, there is mention of fungi being sampled in all plant compartments. I think the bacterial aspects of the work could stand on their own, but I also think it's valuable to have the additional fungal data out in the world. So I would suggest leaving them in and just being a little more comprehensive when explaining the rationale for just focusing on fungi in the one compartment.

Specific comments:

line 183: ITS is not a gene.
lines 210-211: The sequences were removed if the window score dropped below five? Perhaps you mean they were trimmed?
line 220: Why this number of sequences?
line 223: Perhaps this is a typo? It would make sense if it were 280,000
line 239: This is a reasonable cutoff, but then your species accumulation curves are much less informative if they are drawn without these rare OTUs.
line 322: In this paragraph and elsewhere, I think it would be helpful to include the degrees of freedom used for the ANOVA F-test. Since there are so many comparisons being made between and within categories, having the degrees of freedom listed would help to clarify the number of groups and the number of samples being compared.
line 335: Did or did not? Most people generally consider alpha = 0.05 to be a reasonable cutoff, not alpha = 0.1.
line 362: How can you be sure that the effect you observe is due to the range edge and not just simply an increase in elevation (or some other abiotic or biotic factor)?
line 363: But might there also be associational gains? In many cases the diversity is not significantly different even if the community composition is.
line 386: I think this is a bit too strong of a statement. You don't show any evidence that indicates that the microbial communities could be having a causal effect on range edge dynamics.

Figure 3: I would suggest including the legend on this figure -- it would make it easier to interpret at a glance.
Figure 5: It isn't stated in the caption or on the figure which of each pair is within vs edge.

Table 3: I think these p-values should be recalculated with proper restricted permutations. That won't affect the R^2 value, so I imagine the primary trends will be robust, but I do not think that the p values are likely to be accurate.
Table 5: Are these the mean of overall values? vs the means of each of the two elevational classes (within vs edge)? What are the degrees of freedom being used here?

Figure S1: I think a map of the study area would be useful in addition to this diagram. That would help the reader evaluate the proximity of the sites to one another.

Table S1: I think this table could be augmented with some of the data in the 'metadata' table provided on figshare. I'd also be curious about the other factors that could be driving your results -- temperature, surrounding species, substrate changes, etc.

Sources cited:
Fierer, N., C. M. McCain, P. Meir, M. Zimmermann, J. M. Rapp, M. R. Silman, and R. Knight. 2011. Microbes do not follow the elevational diversity patterns of plants and animals. Ecology 92:797-804.

---

## Round 0.2 · accepted · Accept

Thank you for a thoughtful reanalysis of your manuscript. The reviewers and I agree that it is much improved and nearly ready for publication. The reviewer has made some suggestions for minor edits (I second the suggestion for the use of "epiphytes"). Please make these suggestion corrections at the proof stage.

# ·

Basic reporting

Meets all standards.

Experimental design

Meets all standards.

Validity of the findings

Meets all standards.

Additional comments

A much improved manuscript! Very clearly written and results nicely presented.

Three minor things to fix:

Is it 'Acer saccharum Marsh' or 'Acer saccharum Marshall'? They are both used at different places in the manuscript.

The manuscript switches back and forth between ectophytes and epiphytes. I would stick with epiphytes throughout, since it's the term I think more people will be familiar with.

The ordination figure captions say they are PCoA ordinations, but the axis labels on the figures (and the methods section) say NMDS. Which is it?